# The effect of respiratory gases and incubation temperature on early stage embryonic development in sea turtles

**David Terrington Booth**[1]*, **Alexander Archibald-Binge**[1], **Colin James Limpus**[2]

**1** School of Biological Sciences, The University of Queensland, Qld, Australia, **2** Queensland Government Department of Environment and Science, Aquatic Threatened Species Unit, Dutton Park, Qld, Australia

* d.booth@uq.edu.au

**Data Availability Statement:** All relevant data are within the manuscript and its Supporting Information files.

## Abstract

Sea turtle embryos at high-density nesting beaches experience relative high rates of early stage embryo death. One hypothesis to explain this high mortality rate is that there is an increased probability that newly constructed nests are located close to maturing clutches whose metabolising embryos cause low oxygen levels, high carbon dioxide levels, and high temperatures. Although these altered environmental conditions are well tolerated by mature embryos, early stage embryos, i.e. embryos in eggs that have only been incubating for less than a week, may not be as tolerant leading to an increase in their mortality. To test this hypothesis, we incubated newly laid sea turtle eggs over a range of temperatures in different combinations of oxygen and carbon dioxide concentrations and assessed embryo development and death rates. We found that gas mixtures of decreased oxygen and increased carbon dioxide, similar to those found in natural sea turtle nests containing mature embryos, slowed embryonic development but did not influence the mortality rate of early stage embryos. We found incubation temperature had no effect on early embryo mortality but growth rate at 27°C and 34°C was slower than at 30°C and 33°C. Our findings indicate that low oxygen and high carbon dioxide partial pressures are not the cause of the high early stage embryo mortality observed at high-density sea turtle nesting beaches, but there is evidence suggesting high incubation temperatures, particularly above 34°C are harmful. Any management strategies that can increase the spacing between nests or other strategies such as shading or irrigation that reduce sand temperature are likely to increase hatching success at high-density nesting beaches.

## Introduction

The oxygen limitation hypothesis predicts that decreased oxygen availability in the environment and / or limitations in internal oxygen transport can limit aerobic metabolic processes at the intracellular level and thus limit cellular metabolism [1]. In ectothermic animals this potential limitation on cellular process may be exacerbated by high temperatures because high temperatures within the viable temperature range accelerate biochemical reactions and thus oxygen demand, a phenomenon termed the oxygen-capacity-limited thermal tolerance

**Funding:** This project was funded by the The Raine Island Recovery Project which is a five-year, $7.95 million collaboration between BHP, the Queensland Government, the Great Barrier Reef Marine Park Authority, Wuthathi and Kemerkemer Meriam Nation (Ugar, Mer, Erub) Traditional Owners and the Great Barrier Reef Foundation. Grant supplied to DTB. The funders had no role in study design, data collection and analysis, decision to publish, or preparation of the manuscript.

**Competing interests:** The authors have declared that no competing interests exist.

hypothesis [2]. For example, lethal temperature decreased from 44˚C to 40˚C as oxygen decreased from 21% to 10% in a viviparous lizard [3].

Sea turtle embryos experience varying oxygen and carbon dioxide partial pressures and incubation temperatures during their development in the female reproductive tract and natural nests [4, 5]. Embryos begin their development inside the oviduct of the female where the oxygen partial pressure is very low, less than 1 kPa and embryo development arrests at the mid to late gastrula stage due to lack of oxygen [4]. The carbon dioxide partial pressure is probably also considerable higher than in air but this has not been measured. Once the eggs are laid, they are exposed to higher oxygen partial pressures so that the oxygen limitation is lifted and the embryos break developmental arrest about 12 hours after laying and recommence development [4]. However, once developmental arrest is broken, if embryos are re-exposed to extremely low oxygen partial pressures they die of asphyxiation [6].

Typically, during the early stages of incubation, sea turtle embryos experience oxygen and carbon dioxide partial pressures close to those of the atmosphere above the sand [7, 8] and temperatures in the range 28–30˚C and these are considered to be optimal conditions for embryonic development. However, during the second half of incubation, as the embryos grow rapidly in size, the combined metabolism of the entire clutch causes the oxygen partial pressure within the nest to decrease and the carbon dioxide partial pressure to increase [7, 8]. In green turtle (*Chelonian mydas*) nests, during peak metabolism, the oxygen partial pressure can fall to 10 kPa and carbon dioxide partial pressure climb to 8 kPa [7], while in loggerhead turtle (*Caretta caretta*) nests the equivalent values are 15–17 kPa and 3–5 kPa, respectively [7, 8]. In natural nests oxygen and carbon dioxide always change as mirror images of each other, i.e., as oxygen is consumed by the embryo it releases carbon dioxide so that as oxygen decreases within the nest, carbon dioxide increases [7, 8]. The combined metabolism of embryos also generates considerable heat which causes the nest temperature to rise between 2–5˚C above the surrounding sand temperature which is typically 28˚C to 32˚C during the final stages of incubation so maximum nests temperatures can exceed 36˚C [9, 10].

Clutches of eggs laid at high-density sea turtle nesting beaches such as green turtles on Raine Island, Australia, and olive ridley turtles (*Lepidochelys olivacea*) nesting in arribada aggregations in Costa Rica, typically experience much higher mortality of embryos (30–80% mortality) than clutches of eggs laid at low nest density beaches like Heron Island, Australia (0–10% mortality) [11–14]. Much of this mortality occurs due to clutch destruction by subsequent nesting females digging up previously constructed nests during their nesting process [14]. However, even in nests that remain undisturbed throughout incubation, embryo mortality is typically much higher than in nests constructed at low-density nesting beaches, with the majority of embryos dying at a very early stage of development [14].

The physical attributes of the nest environment have been hypothesised as the cause of the elevated early stage embryo mortality [11–14]. Nests constructed at high-density beaches may experience lower oxygen, higher carbon dioxide and higher temperature conditions compared to low nest density beaches [11–14]. Although late stage sea turtle embryos are tolerant to exposure to decreased oxygen, elevated carbon dioxide and elevated temperatures [5, 15, 16], early stage embryos may not be tolerant, possibly because they have not yet developed the ability to induce a heat-shock protein response which has protective effects against environmental induced stress [17]. Hence, if a new clutch of eggs is laid adjacent to a maturing clutch as may frequently occur at high-density nesting beaches, the newly laid eggs may be exposed to low oxygen, high carbon dioxide and elevated temperature conditions which could increase the frequency of early embryo death. Indeed, high early stage embryo mortality has been reported in green turtle nests that were laid near maturing nests, however, it was not possible to determine if the respiratory gas conditions or elevated temperatures or a combination of these two

factors were responsible for the increase in embryo death [14]. In the current study, through a set of controlled incubation experiments, we investigate if incubation conditions that vary significantly from optimal with respect to oxygen and carbon dioxide, elevated temperatures or a combination of these factors increase the frequency of early embryo mortality in sea turtle embryos.

## Methods

### Ethics statement

This study was approved by the University of Queensland NEWMA animal ethics committee (certificate SBS/396/18), and eggs were collected under a scientific purposes permit issued by the Queensland Government National Parks Service (permit PTU18-001406).

### Sources of eggs

Loggerhead turtle (*Caretta cartetta*) eggs were collected (four clutches consisting of 99, 102, 103 and 105 eggs) during oviposition at Mon Repos beach (24.8059˚S, 152.4416˚E) between 4/12/2018 and 15/12/2018 and green turtle (*Chelonia mydas*) eggs were collected (four clutches consisting of 98, 104, 106 and 110 eggs) at Heron Island (23.4423˚S, 151.9148˚E) between 06/01/2019 and 19/01/2019. We performed all experiments at the research facility located within the Mon Repos Conservation Park within 100 m of Mon Repos beach. We transferred eggs from loggerhead turtle nests on Mon Repos beach to the laboratory by hand-carried bucket immediately after collection to be processed. We transferred green turtle eggs immediately after oviposition into an insulated plastic container (60 cm x 35 cm x 35 cm, LxWxH) with its lid open and held overnight in a cool room at 5˚C to 8˚C. This temperature slows down embryo development so that the embryos do not undergo movement-induced mortality during transportation and the hatching rates of eggs transported in this way are similar to untreated eggs [18, 19]. One hour prior to boat departure from Heron Island, the container lid was closed and transported by a 2-hour boat trip to Gladstone followed by a 2.5-hour car trip to Mon Repos where we processed eggs before placing them into incubators, 16 hours after we collected the eggs. We collected two clutches of eggs each night for each species, and we repeated this process once for each species. Hence, in total, we used four clutches of loggerhead turtle eggs and four clutches of green turtle eggs in our experiments.

### Egg incubation

Once at Mon Repos laboratory, we rinsed each egg briefly in distilled water to remove grains of sand, ensuring not to submerge eggs for more than a minute. We weighed eggs and labelled them with a unique identification including clutch and egg number using a 2B pencil. We then placed eggs into a gas tight plastic container (Sistema Klip, 4L, 20cm x 20cm x 10cm) labelled with its temperature and respiratory gas mixture treatment. Each container contained ~8 eggs from one female and ~8 from the other female sampled that night (total number of eggs in each container depended on total number of eggs in a clutch). Within the containers, we buried eggs in sand sourced from Raine Island that had been heat sterilized. We used this sand because Raine Island is host to a high nest-density population of green turtles and many nests experience high early stage embryo mortality at this location [14]. We added distilled water (60g distilled water to 1kg of sand– 6% w/w) to the sand to ensure eggs remained well hydrated during incubation. We placed containers into their designated incubator with unsealed lids exposing eggs to room air for the initial 36 hours of incubation. This procedure insured embryos broke developmental arrest before we exposed eggs to the respiratory gas treatments.

After this initial 36 h period, we examined eggs to check if development had begun as indicated by the appearance of a white-patch on top of the egg. In turtle and crocodile eggs, a white-patch occurs in the eggshell immediately above the embryo 12-36 hours after the start of incubation as fluid is extracted from the eggshell, causing it to dry and turn white [20, 21]. As embryonic development continues, it is thought that dehydration of the albumen below the shell caused by transfer of water from the albumen to the sub-embryonic fluid that surrounds the embryo causes expansion of the white-patch until eventually the white-patch completely surrounds the entire egg [21]. Thus, the rate of expansion of the white-patch is assumed to reflect the rate of embryo development. We assumed that any eggs in which a white-patch was not visible after the initial 36 hours of incubation in air were dead and removed them from the container. We dissected these dead eggs, and examined their contents under a dissecting microscope in order to assign the embryo to a development stage based on morphological criteria described by Miller et al. [22]. In developing eggs, we traced the outside edge of the white-patch using a 2B pencil and photographed them from above.

After the 36 h initiation period, we returned the eggs to their incubators and applied the gas treatments. For loggerhead turtle eggs, 12 treatments were used, a Latin square design of three incubation temperatures, 27˚C, 30˚C and 33˚C, and four respiratory gas mixtures, $O_2$ = 21%, $CO_2$ = 0% (room air = control); $O_2$ = 17%, $CO_2$ = 4%; $O_2$ = 14%, $CO_2$ = 7% and $O_2$ = 10%, $CO_2$ = 11%. Unfortunately, by the time green turtle egg trials were run, the supply of the $O_2$ = 17%, $CO_2$ = 4% gas mixture was exhausted, so this treatment was replaced with an air and 34˚C treatment. We choose the 27˚C, 30˚C and 33˚C temperatures and respiratory gas concentrations as they reflect the range of conditions typically experienced in natural sea turtle nests [7–10]. The ratio of oxygen to carbon dioxide was based on the ratios found in natural green turtle nests on Raine Island [14]. The 34˚C temperature was used because continuous incubation at this temperature is reported to be fatal to sea turtle embryos [5, 23]. Each incubator held a container ventilated at 50ml/min with each of the experimental gas mixtures supplied from premixed gas cylinders. We measured gas concentrations in each container twice a day by aspirating gas leaving the containers into an $O_2$/$CO_2$ analyser (Quantek 970, USA) to check that eggs were exposed to the appropriate gas concentrations. We also checked temperature inside each incubator twice daily using calibrated mercury in glass thermometers. We incubated all eggs for a further 5.5 days after initiation of the gas mixture exposure. Because we were only interested in investigating embryo death during the first week of incubation, we did not continue laboratory incubation of eggs past this point.

After 5.5 days of gas exposure we removed containers from incubators and examined the eggs. If the initial white-patch had not expanded in size, we assumed the embryo was dead and removed the egg, which we dissected and staged. If the white-patch had expanded we assumed the embryo was alive and traced the new white-patch boundary with a 2B pencil and photographed the egg from the side. An estimate of white-patch coverage as a percent of the entire egg surface was made using ImageJ (https://imagej.nih.gov/ij/). The increase in the area of the white patch from 36h to 5.5 days was used as an estimate of growth rate of the embryo. To confirm our assumption that embryos within eggs were alive we selected one egg at random from each treatment to be dissected and staged. We used the white-patch size and developmental stage of embryos of these opened eggs to confirm that white-patch size is correlated with embryo development. Because both loggerhead and green turtles are endangered or threatened species, we did not kill all eggs to ascertain that they were living and to assess their embryonic development stage. We relocated the surviving eggs into artificial nests consisting of 100 to 120 eggs pooled from the different treatments on Mon Repos Beach. Because we could not identify from which treatment hatchlings came from in these artificial nests, we could not

assess the effect of treatment during the first 7 days of incubation on hatching success, so this data is not reported.

## Statistical analysis

We used Pearson correlation to test for a relationship between relative white-patch area and embryo developmental stage. We arcsin transformed relative white-patch area data before ANOVA analysis. We used a mixed model ANOVA (clutch random factor, incubation temperature and respiratory gas treatment fixed factors) for analysis of relative white-patch area for both loggerhead and green turtle eggs. Statistical significance was assumed if $p < 0.05$. We performed all statistical procedures using Statistica© version 13 software.

## Results

### Embryo mortality

Only 9 (5 loggerhead turtle eggs, 4 from one clutch, 1 from another clutch; 4 green turtle eggs, 2 from one clutch and 2 from another clutch) out of 827 eggs (409 loggerhead turtle eggs, 418 green turtle eggs) failed to form a white-patch. Dissections indicated that these eggs were fertile but died in very early stages of development soon after oviposition (developmental stages 6–8 of Miller et al. [22]). For loggerhead eggs that formed a white-patch, with the exception of one embryo incubated at 27˚C in 14% $O_2$, 7% $CO_2$, which died at development stage 9, there was no embryo mortality between the 36 h to 7 day period of gas exposure across all treatments (Table 1). Similarly, mortality of green turtle embryos was very low across all treatments between 36 h and 7 days of incubation (Table 2) and embryos died between developmental stages 10 and 12.

### Embryo development

We assessed embryo development after the 5.5 day exposure to the gas treatments by two methods; direct staging of embryonic development by dissection of one egg from each treatment, and by the relative white-patch area. All embryos were alive as indicated by a beating heart when dissected. Relative white-patch area was correlated with developmental stage in both loggerhead and green turtles (Table 3) indicating that relative white-patch area is a good indicator of developmental stage during early incubation. For this reason, and because the sample size for relative white-patch area was much greater than for developmental stage, only detailed analysis of relative white-patch area is presented. In loggerhead turtle embryos, both incubation temperature and respiratory gas treatment influenced growth of the white-patch, and the interaction terms temperature*gas and clutch*temperature*gas were significant (Table 4). In green turtle embryos, all three factors, clutch, temperature and gas influenced growth of the white-patch, and the interaction terms temperature*gas and clutch*temperature*gas were significant (Table 5). The trend in both species was for growth rate to increase from 27˚C to 30˚C, and then to remain similar between 30˚C and 33˚C, and for green turtle eggs that were incubated at 34˚C in air, growth of the white-patch was slowest at this temperature compared to eggs incubated in air at other temperatures (Fig 1). The rate of growth of the white-patch decreased when the respiratory gases embryos were exposed to became increasingly different from the optimal atmospheric condition of 21% oxygen, 0% carbon dioxide, for both loggerhead and green turtle embryos (Fig 1). While dissecting eggs to record embryo development stage, we found that one green turtle embryo from the 33˚C, 21% $O_2$, 0% $CO_2$ treatment was malformed but alive.

**Table 1.  Experimental treatments used to incubate loggerhead turtle (*Carretta caretta*) eggs and the absolute number as well as the proportion of embryos alive at day 7 after 5.5 days exposure to the gas mixture incubation treatment.**

| Incubation temperature (ºC) | Gas mixture | Number of eggs set | Number of eggs with white-patches at 36 h | Number of embryos alive at 7 days | Proportion survived (%) |
|---|---|---|---|---|---|
| 27 | 21% $O_2$, 0% $CO_2$ | 32 | 31 | 31 | 100 |
| 27 | 17% $O_2$, 4% $CO_2$ | 34 | 33 | 33 | 100 |
| 27 | 14% $O_2$, 7% $CO_2$ | 36 | 36 | 35 | 97 |
| 27 | 10% $O_2$, 11% $CO_2$ | 32 | 32 | 32 | 100 |
| 30 | 21% $O_2$, 0% $CO_2$ | 32 | 32 | 32 | 100 |
| 30 | 17% $O_2$, 4% $CO_2$ | 32 | 32 | 32 | 100 |
| 30 | 14% $O_2$, 7% $CO_2$ | 32 | 31 | 31 | 100 |
| 30 | 10% $O_2$, 11% $CO_2$ | 32 | 32 | 32 | 100 |
| 33 | 21% $O_2$, 0% $CO_2$ | 36 | 35 | 35 | 100 |
| 33 | 17% $O_2$, 4% $CO_2$ | 35 | 35 | 35 | 100 |
| 33 | 14% $O_2$, 7% $CO_2$ | 40 | 39 | 39 | 100 |
| 33 | 10% $O_2$, 11% $CO_2$ | 36 | 36 | 36 | 100 |

**Table 2.  Experimental treatments used to incubate green turtle (*Chelonia mydas*) eggs and the absolute number as well as the proportion of embryos alive at day 7 after 5.5 days exposure to the gas mixture incubation treatment.**

| Incubation temperature (ºC) | Gas mixture | Number of eggs set | Number of eggs with white-patches at 36 h | Number of embryos alive at 7 days | Proportion survived (%) |
|---|---|---|---|---|---|
| 27 | 21% $O_2$, 0% $CO_2$ | 36 | 36 | 36 | 100 |
| 27 | 17% $O_2$, 4% $CO_2$ | 35 | 35 | 35 | 100 |
| 27 | 10% $O_2$, 11% $CO_2$ | 37 | 37 | 37 | 100 |
| 30 | 21% $O_2$, 0% $CO_2$ | 36 | 36 | 36 | 100 |
| 30 | 17% $O_2$, 4% $CO_2$ | 35 | 35 | 35 | 100 |
| 30 | 10% $O_2$, 11% $CO_2$ | 37 | 37 | 36 | 97 |
| 33 | 21% $O_2$, 0% $CO_2$ | 36 | 35 | 35 | 100 |
| 33 | 17% $O_2$, 4% $CO_2$ | 36 | 36 | 36 | 100 |
| 33 | 10% $O_2$, 11% $CO_2$ | 36 | 35 | 33 | 94 |
| 34 | 21% $O_2$, 0% $CO_2$ | 94 | 92 | 89 | 97 |

**Table 3. Data relating the embryonic development stage (Miller et al. [22]) to the relative white-patch area for loggerhead and green turtle embryos after 7 days of incubation.**

| Loggerhead turtle | | | |
|---|---|---|---|
| Incubation temperature (°C) | Gas mixture | Developmental stage | Percent of shell covered by the white-patch |
| 27 | 21% $O_2$, 0% $CO_2$ | 13 | 88 |
| 27 | 17% $O_2$, 4% $CO_2$ | 13 | 62 |
| 27 | 14% $O_2$, 7% $CO_2$ | 10 | 49 |
| 27 | 10% $O_2$, 11% $CO_2$ | 10 | 34 |
| 30 | 21% $O_2$, 0% $CO_2$ | 14 | 83 |
| 30 | 17% $O_2$, 4% $CO_2$ | 18 | 71 |
| 30 | 14% $O_2$, 7% $CO_2$ | 15 | 58 |
| 30 | 10% $O_2$, 11% $CO_2$ | 13 | 45 |
| 33 | 21% $O_2$, 0% $CO_2$ | 16 | 84 |
| 33 | 17% $O_2$, 4% $CO_2$ | 17 | 69 |
| 33 | 14% $O_2$, 7% $CO_2$ | 15 | 60 |
| 33 | 10% $O_2$, 11% $CO_2$ | 16 | 66 |
| Pearson correlation | $r^2 = 0.33$, t = 2.21 | | |
| | p = 0.049, n = 12 | | |
| Green turtle | | | |
| Incubation temperature (°C) | Gas mixture | Developmental stage | Percent of shell covered by the white patch |
| 27 | 21% $O_2$, 0% $CO_2$ | 16 | 84 |
| 27 | 17% $O_2$, 4% $CO_2$ | 15 | 68 |
| 27 | 10% $O_2$, 11% $CO_2$ | 15 | 37 |
| 30 | 21% $O_2$, 0% $CO_2$ | 18 | 85 |
| 30 | 17% $O_2$, 4% $CO_2$ | 19 | 82 |
| 30 | 10% $O_2$, 11% $CO_2$ | 13 | 43 |
| 33 | 21% $O_2$, 0% $CO_2$ | 16 | 77 |
| 33 | 17% $O_2$, 4% $CO_2$ | 19 | 81 |
| 33 | 10% $O_2$, 11% $CO_2$ | 12 | 48 |
| Pearson correlation | $r^2 = 0.62$, t = 3.34 p = 0.012, n = 9 | | |

# Discussion

## Respiratory gases

Contrary to our expectation, exposure of early stage embryos to low oxygen and high carbon dioxide gas mixtures did not influence mortality during the first week of incubation. Hence,

**Table 4. Mixed factor ANOVA table for loggerhead turtle (*Carretta carretta*) egg arcsin transformed relative white-patch area after 7 days exposure to incubation temperature, and 5.5 days exposure to gas mixtures.**

| Effect | SS | DF | MS | F | P |
|---|---|---|---|---|---|
| Intercept | 208.5 | 1 | 208.5 | 394.2 | < 0.001 |
| Clutch (random) | 1.6 | 3 | 0.5 | 25.0 | 0.420 |
| Incubation temperature (fixed) | 1.2 | 2 | 0.6 | 11.2 | 0.009 |
| Respiratory gas (fixed) | 15.1 | 3 | 5.0 | 95.9 | < 0.001 |
| Clutch*Temperature (random) | 0.3 | 6 | 0.1 | 0.6 | 0.698 |
| Clutch*Gas (random) | 0.5 | 9 | 0.1 | 0.6 | 0.777 |
| Temperature*Gas (fixed) | 2.2 | 6 | 0.37 | 4.3 | 0.007 |
| Clutch*Temperature*Gas (random) | 1.6 | 18 | 0.1 | 6.6 | < 0.001 |
| Error | 4.6 | 352 | 0.01 | | |

**Table 5. Mixed factor ANOVA table for green turtle (*Chelonia mydas*) egg arcsin transformed relative white-patch area after 7 days exposure to incubation temperature, and 5.5 days exposure to gas mixtures.**

| Effect | SS | DF | MS | F | P |
|---|---|---|---|---|---|
| Intercept | 189.8 | 1 | 189.9 | 186.1 | < 0.001 |
| Clutch (random) | 3.1 | 3 | 1.0 | 17.8 | 0.003 |
| Incubation temperature (fixed) | 0.6 | 2 | 0.3 | 12.7 | 0.007 |
| Respiratory gas (fixed) | 17.5 | 2 | 8.7 | 169.1 | < 0.001 |
| Clutch*Temperature (random) | 0.2 | 6 | 0.1 | 1.3 | 0.335 |
| Clutch*Gas (random) | 0.3 | 6 | 0.1 | 2.7 | 0.071 |
| Temperature*Gas (fixed) | 1.1 | 4 | 0.27 | 14.0 | < 0.001 |
| Clutch*Temperature*Gas (random) | 0.2 | 12 | 0.1 | 1.8 | 0.046 |
| Error | 3.0 | 277 | 0.01 | | |

we reject the respiratory gas hypothesis as an explanation of early embryo death syndrome so prevalent in high nest-density sea turtle rookeries. However, we found that as the respiratory gases deviated further from atmospheric air, the rate of embryo development decreased, so clearly there was a deleterious effect of exposure to respiratory gases that deviate from atmospheric conditions at this early stage of embryonic development. If new clutches are laid in close proximity to maturing clutches, this can potentially cause the respiratory gases in the newly laid clutch to deviate significantly from atmospheric air. Under these conditions, the clutch may experience a longer incubation period than other clutches that are not influenced by altered respiratory gases. However, in nature, clutches are only exposed to self-generated low oxygen and high carbon dioxide partial pressures during the last 2 weeks of incubation. After this time, hatchlings escape the nest and oxygen and carbon dioxide partial pressures return to atmospheric conditions. So, even if a new clutch was laid next to a late maturing clutch, embryos in the new clutch are most likely only exposed to adverse respiratory gases for a maximum of two weeks unless they also have multiple clutches of eggs with embryos at different developmental stages next to them.

## Incubation temperature

Although incubation temperature appeared to have little effect on embryo mortality over the first 7 days of incubation, continuous incubation at 34˚C from the beginning of incubation results in death of all sea turtle embryos [5, 23–25]. The observation that an embryo was still alive, but malformed after 7 days of incubation at 33˚C suggests it is possible that even though embryos incubated at this temperature are still alive after a week of incubation, some embryos have developed high temperature induced teratogenic malformations that will result in embryo death later in incubation. Hence, high nest temperatures experienced early in incubation is a possible explanation for the large number of nests experiencing a high rate of early embryo death in high-density nesting aggregations. Indeed, a natural green turtle clutch that experienced 80% early embryo death at Raine Island experienced an average temperature of 35.5˚C during its first week of incubation [14]. In natural sea turtle nests, a rise in nest temperature due to metabolic heating is always accompanied by a fall in oxygen and increase in carbon dioxide, and it is generally theorised that a decrease in oxygen would exacerbate the detrimental effect of high temperature [2], and this has been indicated in developing lizard embryos [3]. However, we found no evidence of this exacerbated effect in our early stage sea turtle embryos, with high survivability during the first week of incubation when exposed to hypoxia in combination with hypercapnia across all experimental temperatures.

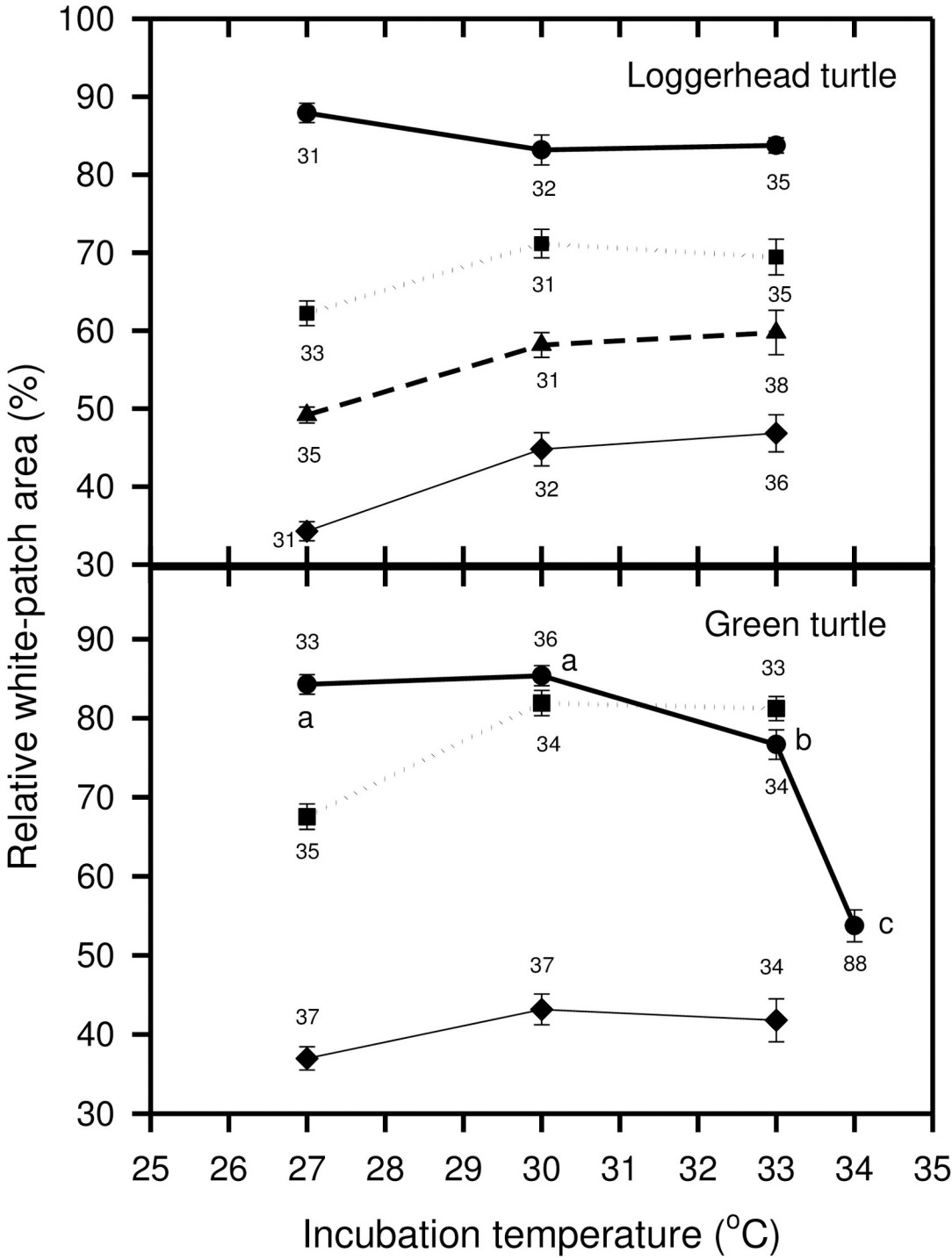

**Fig 1. Plot of relative white-patch size on day 7 after 5.5 days of exposure to different gas mixes at different incubation temperatures for loggerhead (*Carretta carretta*) and green turtle (*Chelonia mydas*) eggs.** Thick solid line and circles = 21% $O_2$, 0%$CO_2$, dotted line and squares = 17%$O_2$, 4%$CO_2$, dashed line and triangles = 14%$O_2$, 7%$CO_2$, thin solid line and diamonds = 10%$O_2$, 11%$CO_2$. Error bars = Standard errors. Numbers associated with symbols = number of eggs in sample. Letters adjacent to symbols for the 21%$O_2$, 0%$CO_2$ gas treatment for green turtle eggs indicate significant differences according to a Tukey post-hoc test adjusted for unequal sample sizes.

Although high incubation temperatures were not immediately fatal during the first week of incubation, they did retard the growth of early stage embryos. In ectotherms, including sea turtles, embryonic growth and development rate increase with an increase in incubation temperature within the viable temperature range. This explains the increase in embryo development rate we observed between 27˚C and 30˚C, however development rate did not continue to increase at 33˚C, and actually decreased at 34˚C in green turtle eggs when eggs were incubated in air. It would appear that incubation at temperatures of 33˚C and higher are sub-optimal and that although development can continue, these high temperatures may cause cellular damage that needs to be repaired, and this damage slows the rate of development. Alternatively, slowing of development at high temperatures may be a tactic to prevent high temperature damage, by waiting until cooler temperatures return before recommencing development. Ultimately, for early stage embryos incubated at temperatures of 34˚C and higher for long periods, the high temperature results in embryo death [5, 23–25].

We collected the eggs for our experiments from low density nesting populations in which we knew by checking flipper tag numbers that the females laid their clutch on their first laying attempt. Previous reports [26–28] indicate that retention of eggs for several days within the female reproductive tract due to multiple failed nesting attempts that frequently occurs at high nest-density beaches, causes a decrease in egg viability. Hence, it is possible that at high nest-density beaches a combination of prolonged egg retention and exposure of eggs to low oxygen, high carbon dioxide partial pressures and high nest temperatures result in a higher frequency of early stage embryo mortality.

In summary, although we found the development rate of early stage sea turtle embryos is retarded when exposed to partial pressures of oxygen and carbon dioxide typically encountered in maturing sea turtle clutches, such exposure is not fatal. Likewise, when we exposed early stage sea turtle embryos to incubation temperatures of 33˚C and above, development was slowed. At nesting beaches that experience high-density nesting, the close proximity of nests means that many newly laid clutches could experience high temperatures due to the metabolic heat production of nearby maturing clutches, and this might result in elevated rates of early embryo death, especially if temperatures exceed 34˚C for long periods of time. Hence, as the number of nesting females increases at high-density rookeries, the number of newly laid clutches exposed to high temperatures increases, and therefore the proportion of clutches experiencing early embryo death syndrome increases as reported for green turtle clutches at Raine Island [14]. A management strategy that could mitigate this increase in early embryo death could be to increase the beach area suitable for nesting so that nests could be spaced further apart, or the use of other strategies such as shading or irrigation that reduce sand temperature.

## Supporting information

**S1 Dataset.**
(XLSX)

## Acknowledgments

Experimental facilities were provided by the Queensland Government Department of Environment and Science.

## Author Contributions

**Conceptualization:** David Terrington Booth, Colin James Limpus.

**Data curation:** Alexander Archibald-Binge.

**Formal analysis:** David Terrington Booth, Alexander Archibald-Binge.

**Funding acquisition:** David Terrington Booth.

**Investigation:** Alexander Archibald-Binge.

**Methodology:** David Terrington Booth, Colin James Limpus.

**Project administration:** David Terrington Booth.

**Resources:** Colin James Limpus.

**Supervision:** David Terrington Booth, Colin James Limpus.

**Writing – original draft:** David Terrington Booth.

**Writing – review & editing:** Colin James Limpus.

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
