## [Decision Letter · Decision Letter 0]

24 Jul 2020

PONE-D-20-13033

The effect of respiratory gases and incubation temperature on early stage embryonic development in sea turtles

PLOS ONE

Dear Dr. Booth,

Thank you for submitting your manuscript to PLOS ONE. After careful consideration, we feel that it has merit but does not fully meet PLOS ONE’s publication criteria as it currently stands. Therefore, we invite you to submit a revised version of the manuscript that addresses the points raised during the review process.

We look forward to receiving your revised manuscript.

Kind regards,

Frank Melzner

Academic Editor

PLOS ONE

Journal Requirements:

Reviewers' comments:

Reviewer's Responses to Questions

**Comments to the Author**

1. Is the manuscript technically sound, and do the data support the conclusions?

Reviewer #1: Yes

Reviewer #2: Partly

2. Has the statistical analysis been performed appropriately and rigorously? 

Reviewer #1: Yes

Reviewer #2: No

3. Have the authors made all data underlying the findings in their manuscript fully available?

Reviewer #1: No

Reviewer #2: Yes

4. Is the manuscript presented in an intelligible fashion and written in standard English?

Reviewer #1: Yes

Reviewer #2: Yes

5. Review Comments to the Author

Reviewer #1: General points:

The manuscript presents a sound study that answers an important biological question with real conservation consequences for threatened sea turtle populations. I congratulate the authors on designing and executing a valuable study to elucidate the causes of abnormally high early-stage embryonic death at high-density sea turtle nesting beaches.

My main criticism with this manuscript is that, for the two clutches where data was presented for stage of embryonic mortality, the temperature treatments were applied over the entire incubation period and therefore does not reflect a "short-term" early incubation effect of a nearby mature clutch incubating. In the clutches where high temperatures were only applied for the initial period of incubation the authors have not presented the data for stage of embryonic development at which the embryos died. I suggest that the authors either revise the manuscript to include the stage of death data for the unhatched eggs from clutches that were reburied in artificial nests. If these data do not exist, the authors need to address this limitation and I suggest they include a caveat in the discussion to highlight the limitation that embryonic death cannot be conclusively attributed to "short-term early-stage" high incubation temperature. This is the only major change I recommend making to the manuscript, the rest of the points I raise should be easy to address.

What about the impact of extended pre-ovipositional embryonic arrest on embryos ability to survive high nest temperatures and low oxygen partial pressure? At high-density nesting beaches, such as Raine Island and Ostional, it could be likely that there are higher rates of failed nesting attempts due to disturbance from other females. This would mean a higher proportion of females return to nest over subsequent days to attempt oviposition when compared with those nesting at low-density nesting beaches. Inherently, this would mean that embryos are maintained in embryonic arrest for longer periods of time. Previous experiments have found that sea turtle embryos have limited tolerance to remain in embryonic arrest (Rings et al. 2015 Phys. Bio. Zool. and Williamson et al. 2019 Sci. Rep.) and in one population of leatherback turtles longer interesting intervals result in higher proportions of early-stage embryonic mortality (Rafferty et al. 2011 PLoS One). If embryos are potentially compromised by extended arrest they may be more susceptible to higher incubation temperatures and lower oxygen availability when they do recommence development after the arrest. It would be good for the authors to include some acknowledgement of this possibility in the discussion.

Specific points:

Page 1, Line 14: Change to "high death rate" or "high mortality rate".

Page 1, Line 22: Change "sea turtle nest" to "sea turtle nests".

Pages 1 - 2, Lines 23 - 24: Change "did not influence embryo mortality of early stage embryos" to "did not influence the mortality rate of early stage embryos".

Page 2, Line 25: I am unsure what the current word count for the abstract is, but I suggest including a final sentence summarizing the importance and implications of the findings.

Page 2 Line 34: Delete "of" from "increases of lizard"

Page 2, Lines 36-37: Provide a reference for this sentence. Perhaps one of the Ackerman papers on the topic. I suggest also breaking this paragraph into two separate paragraphs. One on the information about developmental arrest and the other on the information about varying gas concentrations and temperature throughout incubation in the nest.

Page 2, Line 39: Change "stalls" to "arrests".

Page 2, Lines 39 - 40: Embryos do not break developmental arrest immediately after oviposition. The study that the authors cite for this sentence suggests that embryos break the arrest after at least 12 hours of exposure to higher oxygen partial pressures. Please change this sentence to reflect this difference.

Page 2, Line 42: Change "development arrest" to "developmental arrest".

Page 3, Line 53: Change "it released carbon dioxide" to "it releases carbon dioxide".

Page 3, Line 57: Change “aggregations such as” to “aggregation beaches, such as”.

Page 3, Line 58: Arribadas is plural in Spanish. Change "nesting in arribadas aggregations" to "nesting in arribadas" or "nesting in arribada aggregations"

Page 3, Line 68: Again, this paragraph is very long. Suggestion to split into two paragraphs after this sentence.

Page 3, Lines 70-72: Provide appropriate references for this section of the sentence. Possibly Nicki Mitchell et al papers on heat-shock proteins in sea turtle embryos e.g. Bentley et al. 2015 Mol. Ecol.; or Tedeschi et al 2015 or 2016.

Page 5, Line 111: Please provide more information on the type of containers used for the application of the gas treatments.

Page 5, Lines 113 - 114: Unless I have missed it, there is no explanation provided for the use of Raine Island sand during incubation. Please explain the reasoning for this.

Page 6, Line 131: Provide reference to the staging criteria that you staged embryos with. I assume Miller's 1985 staging guide?

Page 7, Line 159: Please provide clarification whether these eggs were maintained at their assigned temperature treatment or if all eggs were then incubated at a similar temperature.

Page 7, Line 160: Please explain a little more about what happened to these eggs. Were they opened and staged to assess viability? Were the eggs left to incubate until hatching in the incubators?

Page 8, Lines 188-191: I assume you excavated and staged the unhatched eggs from the clutches that were reburied in artificial nests as well. You should provide a summary of these data also. Or explain why this was not done. If the primary concern of this study is early stage death I think it is quite important to present the data on which stage of development these embryos died at.

Page 9, Line 198: This is an important finding and indicates that growth of the white spot is actively regulated by the embryo rather than passively by desiccation or other environmental influences.

Page 10, Line 228: Change "logger turtle" to "loggerhead turtle"

Page 10, Lines 235-237: It is important the authors have acknowledged this potential limitation in assessing the difference in growth rates between the two species given that they applied different methodologies to the eggs dependent on species.

Page 12, line 274: Change "early stage sea turtle embryos development rate" to "the development rate of early

stage sea turtle embryos".

Page 19, Lines 426 – 427 and Page 20, Lines 430 - 431: Table 1 & 2 Captions are nonsensical to me. The experimental treatments cannot indicate survival in themselves. I suggest changing "turtle eggs indicating survival" to "turtle eggs and embryo survival" or something similar.

Reviewer #2: Generally, I found the study by Booth et al. interesting and relevant from a conservation perspective and the collected data is suitable to answer the main study question. The manuscript itself is concise, well-written with appropriate referencing. However, my major concern is that the experimental design is unclear and does not always seems to address the main study question, which could be amended by a careful revision of the M&M section. In addition the authors draw conclusions that are not supported by their data or their experimental approach: i) by comparing the two study species, which is not supported by statistics or a comparable handling of the embryos, ii) by concluding that development is affected by hypoxia, which was not tested independently from hypercapnia and iii) by confounding the results from a short-term exposure in a common garden experiment with the results from a long term survival trial (where embryos were pre-exposed to different gas tensions, but then kept in ambient air). I think all of these concerns can be addressed, but will require re-analyzing some of the data, changing the presentation of the results and re-writing large parts of the discussion to adhere more closely to the study question and the data at hand. Therefore, I cannot recommend the manuscript for publication in PlosOne in its current form and in the following I list 5 major concerns that I would like the authors to address. In addition, I have listed a number of minor comments that will hopefully help with the revision of this manuscript.

1. The discussion is well-written and meaningful as far as it pertains to the main research question. But to a large extent the authors take the discussion beyond what can be concluded from their data to an extent that is unacceptable for publication. I recommend the authors adhere closely to the main research question and avoid over-interpreting their data. For instance , the authors conclude that one species is more susceptible to hypoxia than the other, based on a metric of developmental rate (L227-245), but this isn’t supported by any statistical analysis, the embryos were not treated in the same way prior to experiments and generally potential interspecific differences in developmental rate are not considered. Furthermore, the authors discuss potential effects of hypoxia on development, but hypoxia was always tested in combination with hypercarbia and therefore such mechanistic distinctions cannot be made (L238-245). I urge the authors to fundamentally revise the interpretation of their data and pay close attention to the limitations in their experimental design.

2. I think the experimental design is appropriate to answer the main study question on whether early embryonic development/survival is affected by the conditions (gas and temperature) created by nearby clutches on crowded beaches. This was assessed in a first experiment over 7 days where embryos were exposed to different combinations of gas tensions and temperature and development was monitored. This experiment answers the main study questions and there were no effects on early embryo survival, but development was delayed. After this, things get more confusing and the further experiments are not in line with this initial hypothesis. The second experiment looked at long-term survival of the embryos at different temperatures, but at ambient gas tension (and using the embryos that had been previously exposed to different gas tensions). Here the authors report significant differences in survival, but it’s unclear whether this longer time-frame addresses the initial question about “early embryo mortality” (a term that should be clearly defined early in the paper) and fails to address the actual conditions on crowded beaches that always include altered gas tensions. The presentation and discussion of the results intermixes the findings from these two experiments and it is often unclear which condition is being discussed. I suggest the authors outline more clearly the justification for this second experiment and how it fits within the main study objective. One option would be to re-analyze the data and test whether an early exposure of the embryos to altered gas tensions had an effect on long-term survival even at atmospheric gas tensions (currently the results from all gas-tension pre-exposures are pooled across the temperature treatments). The findings of this analysis would be interesting regardless of the outcome and could tie the two experiments together to test a common hypothesis.

3. In general, the way the authors have described the experimental design of their study is convoluted and even after reading the manuscript again it is not entirely clear to me why certain conditions were tested. I recommend that the authors include a dedicated section in the M&M on experimental design, where they lay out clearly what the experimental treatments were, when each treatment was sampled and how the treatments were replicated (the authors use eggs as N whereas really each clutch should be an experimental unit, where N=4). I also recommend dividing the experiments into clearly defined stages (phases or series). The first from 0-36 h after which measurements were taken, the second 36 h- 7d which tested the effects of gas tensions and temperature on development and finally a long-term exposure 7-46 d in which only survival was assessed.

4. I recommend the authors revisit the presentation of their data. Figures 2 and 3 show both species in one graph, whereas gas tensions were pooled for each temperature. I don’t think this is a valid presentation of the data as there were significant differences between the gas tensions and therefore, they shouldn’t be pooled in the first place. In addition, there was a significant interaction effect between gas tensions and temperature and therefore all combinations of temp*gas need to be shown separately. I recommend showing the p-values of main effects and interactions in the figure as well, as this reflects the outcome of the statistical analysis and Tables 3 and 4 could be omitted. And finally, I suggest presenting one figure per species, as no statistical comparisons are made between the two species.

5. In one of the treatments the pre-mixed gases ran out and the authors chose to convert this treatment into a high-temperature exposure (as far as I understood this section) where embryos were exposed to 34°C instead of 33°C. It is not clear why this additional increase in 1°C was chosen and, without any context, it is surprising that it had large effects on animal survival. I recommend the authors justify their approach more clearly. Since the other gas tensions were pooled by temperature this 34°C exposure is the only treatment that received ambient air throughout development (correct?) and therefore is not necessarily comparable to the other treatments. One way to avoid this confounding effect of pre-exposure to different gas tensions would be to compare the 34°C exposure only to the other treatments that were exposed to ambient air throughout (controls). I suspect the results would still hold up and this would make for a more meaningful comparison (if the results don’t hold up this is reason for concern in itself).

Specific comments:

Abstract

L13-14 Here and in the introduction, please define the term “early stage embryo death”. This is especially important since the study assessed embryo development over two distinct time-frames and throughout the manuscript it is unclear whether both parts of the study address the issue of the early mortality of embryos.

I find the last sentence of the abstract a bit cryptic and, based on this, I was unsure whether there was a clear effect of temperature on embryo mortality. Consider rephrasing this sentence for clarity. Also, I’d like to see a concluding statement in the abstract that is based on the current findings. What does it all mean for wild turtles that nest in crowded beaches?

Introduction

L34 delete “of”. Also, it’s worthwhile mentioning that these were embryos of live-bearing lizards and I would note the level of hypoxia and the temperatures, as well.

L45 replace that” with “those”

L49 I would not use the term peak metabolism, because it implies that metabolism would decrease again throughout embryonic development. Presumably this refers to the last stage in development before hatch.

L53 “releases”

L51-54 I agree that hypoxia and hypercarbia typically go hand-in-hand in natural settings, but the effects on the organisms differ greatly between these two conditions. Overall, I found the paper focused only on O2 whereas the experimental treatments manipulated both gases. I suggest the authors introduce briefly the effects of hypercapnia on embryonic development as well; many good physiological studies have recently dealt with this.

L54-54 What are typical temperatures in the sand? Provide some context so the reader knows if we’re talking about a 30% (sand temp ~10°C) increase in temperature or a 10% increase (sand temp ~30°C).

L56 final stages of incubation

L57-60 I found this sentence rather vague, but it is a critical justification for conducting this study. Please state specifically whether within one species crowded nesting beached have higher embryo mortality rates compered to empty beaches. Be specific and mention how large of an effect this is for each species. As written, it is unclear whether comparisons are made across the two species or these very different habitats.

L72 Are there any studies that indicate an inability of young embryos to mount a heat-shock protein response? Please cite.

L72-79 This is a repetition of earlier statements and I suggest integrating it with previous mentions.

L79 I liked that the authors put forward a testable hypothesis in the abstract and I suggest doing the same here, in the introduction. I also recommend that the authors specifically introduce the conditions that are considered “optimal” for embryo development and how they are going to be manipulated in this study (i.e. replace “suboptimal”).

Materials and Methods

L90-106 In this paragraph, mention the numbers of eggs that were collected. I was surprised to read in the results that there were several hundred eggs per clutch. If anything, this strengthens the paper.

L97 I would avoid mentioning brand names. An insulated cooler or container.

L95-106 I understand that there were some logistical challenges to getting the green turtle eggs into the lab. Presumably the careful transport of these embryos did not affect their development, but it would be good if the authors could justify this in a sentence or two. Were mortality rates generally comparable to clutches that were left undisturbed? Similar to those of the loggerhead turtles that did not experience the transport procedure? Or were they typical for what has been reported for this species in the past?

L136-139 I think the authors should provide a justification for why exactly these gas tensions were used. I understand that they span the conditions that turtles may experience in the wild, but why were exactly these combinations of O2/CO2 chosen? I.e. O2 is reduced in steps of 80%, 66% and 47% relative to normoxia, whereas CO2 is increased in steps of 4%, 7% and 11%. What’s the basis for choosing these ratios? Also, is the 21%/0% treatment really the best control, as wild turtles would probably never experience this in the sand. I’m not saying that there is something fundamentally wrong with these experimental conditions, but I think the authors need to justify specifically why they were chosen.

L149-160 This section is a bit confusing as written and the authors should state more clearly what measurements were performed at which times? How do they know if the embryos grew (and were alive) before tracing the patch and comparing it to the previous measurement?

It seems that two different methods were used to assess whether an embryo was alive, tracing a white spot on the egg and staging embryos by dissection. Why this redundancy is there a reason to think that one method would yield different or unreliable results? And if so, why not use the more reliable of the two methods. And what exactly was assessed during staging of the embryos?

It’s also not clear to me why the embryos were only exposed to the gas mixtures for 7 days. And it sounds like some embryos were returned to their natural nests thereafter, but others were not. It’s not entirely clear to me how this helped addressing the research question. Please clarify.

L157-160 The authors mentioned earlier that one of the gas mixtures ran out and presumably this is why the last clutch of green turtle eggs was kept at ambient gas concentrations throughout, but instead temperature was manipulated, right? There is no mention of that here, which I found confusing.

L159-160 Include, how many eggs were sampled at each timepoint.

Results

L178 The authors have not introduced the metric of developmental stages or how they were assessed. This should be described in the M&M section.

L187-188 It’s still not clear to me exactly how these long-term experiments were performed. It sounds like all loggerhead eggs were returned to the wild after the 7 day exposure, but the green turtle eggs went back into the incubators. At this point gas tensions were ambient for all treatments (L159), but in figure 1 it says that data was pooled across gas treatments. And that we have one additional treatment that was only exposed to air throughout the experiment (because the gas ran out) and this treatment was kept at 34°C instead of 33°C (L138-140). I had to go back and forth several times to piece this information together and I’m still not sure whether this is all correct. I’m also unsure whether the comparisons that are being made are actually meaningful (see major comment about including a dedicated M&M section on experimental design).

L199-200 It’s ok to present only these data, but mention in a sentence whether the two metrics of embryo development agree with one-another or not. Perhaps the staging data could be made accessible in a supplement, to avoid cherry-picking one dataset over the other?

L205-207 This sentence is confusing as written, please re-phrase.

Tables

In general, I would avoid the vertical lines in tables for publication.

In the table captions I would mention the full species names, instead of just “loggerhead or green eggs”.

Figure 1 I would avoid using the thick line in the 34°C treatment as is covers the actual datapoints. Chose a different linetype that is less occlusive. In my version, the R2 appeared as a box. If the data were pooled across gas treatments, I’d like to see some measure of variability included around the means (SD or SEM).

Were there any significant differences in survival between the embryos that had previously been exposed to the different gas tensions? If not mention this here as it justifies the pooling of these data. Otherwise report the result as it would be interesting to know whether hypoxia or hypocapnia during early development has an impact on long-term survival rates.

Figure 2 Since there were significant effects of gas tensions on embryo development these treatment should not be pooled. The presentation of these data will need to be changed. The figure will need to show the development for each gas tension individually and to highlight that there was a significant effect of temperature the authors could include the p-values for the main effects and the interaction term in the figure as well, which would replace Table 3.

The same considerations apply to Figure 3. I recommend the authors revisit the presentation of their data. I suggest presenting one figure per species, as no statistical comparisons are made between the two species. And each of these figures will need to show all combinations of experimental treatments as they were significantly different and there was a significant interaction between gas treatments and temperature, in each case. A more common way to express changes in development with temperature would be to calculate the Q10 values for each step. This is somewhat of a personal preference, but the authors should consider this type of analysis.

Discussion

L217 the gases were not “sub-atmospheric” for CO2

L222-226 I disagree with this statement as it neglects the possibility that there could be a number of adjacent clutches that are staggered in their developmental stage and therefore a single clutch of eggs may experience hypoxia and hypercapnia throughout development or for any fraction of this time.

L227-230 I’m not sure this comparison is backed by the data and no statistical evidence is provided to support this. In addition, differences in the development of a “white patch”, a rather subjective marker, could simply be due to differences between egg size, egg-shell composition, etc. between the species. Based on the provided data it cannot be concluded that one species is more susceptible to the altered gas tensions than the other. This applies to the entire following paragraph from L227-245.

L235-238 I think it’s great that the authors discuss critically the possibility that cooling the embryos of one species before the trials may have confounded the results. But I still think it would be good to dissipate these concerns earlier in the M&M section by providing a sound justification of this approach. I understand that field conditions are usually a compromise and if it was simply not possible to treat the eggs from both species in the same way that can be sated as well. Based on this discrepancy I would further discourage the authors from drawing any conclusions by comparing the development of the two species. The experimental design simply does not support these comparisons.

L238-245 The authors focus on an O2 limitation as the probable cause for an impaired development. Also this cannot be concluded from the data, as O2 and CO2 were always altered in concert. I urge the authors to fundamentally revise the interpretation of their data and pay close attention to the limitations in their experimental design. I think the experimental design is appropriate to answer the main study question on whether embryonic development is affected by the conditions (gas and temperature) of nearby clutches on crowded beaches. I recommend the authors adhere closely to this question and avoid overinterpreting their data. The effects of O2 and CO2 on development of embryos can be discussed and some speculation is ok to suggest future experiments, but the effect of these gases cannot be disentangled unambiguously.

L261-263 Again, based on the presented data I’d like to know whether early hypoxia or hypercapnia had an effect on late embryo mortality in ambient gas tensions. Did the authors test for such an effect? Whatever the finding, this would be interesting to report in the results.

L269-272 I’m not convince that the slower development at higher temperatures (by itself) is an indication of cellular damage or morbidity. In fact, it could be an adaptive response of the embryos to prevent a further deterioration of the conditions surrounding the clutch and perhaps to allow for neighboring clutched to hatch first by delaying their own development. The only data that really suggests that the embryos may be pushed beyond their physiological tolerance is the survival data. I suggest the authors make this distinction more clearly.

L276 “clutches”

L278 at 34°C long-term survival was affected. Again, here the results of the short-term and long-term experiments are intermixed. I recommend differentiating more clearly between effects on development (within the first 8.5 days) and long-term survival, which was affected by temperature.

L281-288 I like this last paragraph that interprets the major findings in an applied context and provides clear management strategies for turtle conservation. I suggest ending the abstract in a similar way.

6. PLOS authors have the option to publish the peer review history of their article (what does this mean?). If published, this will include your full peer review and any attached files.

Reviewer #1: **Yes: **Sean Williamson

Reviewer #2: No

---

## [Author Response · Author response to Decision Letter 0]

3 Sep 2020

All issues raised by the reviewers have been addressed in the response to reviewer's document.

---

## [Decision Letter · Decision Letter 1]

29 Sep 2020

PONE-D-20-13033R1

The effect of respiratory gases and incubation temperature on early stage embryonic development in sea turtles

PLOS ONE

Dear Dr. Booth,

Thank you for submitting your manuscript to PLOS ONE. After careful consideration, we feel that it has merit but does not fully meet PLOS ONE’s publication criteria as it currently stands. Therefore, we invite you to submit a revised version of the manuscript that addresses the points raised during the review process.

While both reviewers acknowledge substantial improvements in the revised version of this ms, they also feel that mortality data should be supplied, along with a critical discusssion of mortality observed in this vs. older studies. I agree with their assessment and suggest you provide the required data and engage in a more critical discussion of mortality in relation to temperature.

We look forward to receiving your revised manuscript.

Kind regards,

Frank Melzner

Academic Editor

PLOS ONE

Reviewers' comments:

Reviewer's Responses to Questions

**Comments to the Author**

1. If the authors have adequately addressed your comments raised in a previous round of review and you feel that this manuscript is now acceptable for publication, you may indicate that here to bypass the “Comments to the Author” section, enter your conflict of interest statement in the “Confidential to Editor” section, and submit your "Accept" recommendation.

Reviewer #1: (No Response)

Reviewer #2: (No Response)

2. Is the manuscript technically sound, and do the data support the conclusions?

Reviewer #1: Partly

Reviewer #2: Partly

3. Has the statistical analysis been performed appropriately and rigorously? 

Reviewer #1: Yes

Reviewer #2: Yes

4. Have the authors made all data underlying the findings in their manuscript fully available?

Reviewer #1: Yes

Reviewer #2: Yes

5. Is the manuscript presented in an intelligible fashion and written in standard English?

Reviewer #1: Yes

Reviewer #2: Yes

6. Review Comments to the Author

Reviewer #1: I am concerned that my main criticism of this manuscript has still not been fully addressed. The study aim is to assess the impact of changes in suboptimal gas mixtures and temperatures during early-embryonic development to ascertain whether this could explain high rates of early embryonic death at high-density nesting beaches. Data on stage of embryonic death has not been presented for sufficient comparison of differences between the treatments. Furthermore, in this revised version, there is no longer any substantial mortality data included. This makes it difficult to definitively conclude that any of the treatments applied caused higher rates of early-embryonic death and poorer hatching success in general. The authors did apparently open 2 eggs (or potentially 4?) from each treatment to assess stage of embryonic development at the end of the treatment but this data has not been reported in the manuscript.

Given the above, if data on stage of embryonic death cannot be provided, the authors need to temper interpretation of results that temps above 34℃ are more harmful than non-optimal gas concentrations. There was minimal difference in the proportion of embryos that you assumed were still alive at 7 days across all treatments, including temperature treatments. Furthermore, the data you report on relative growth of the white-patch indicate that non-ideal gas mixtures (10%O2, 11%CO2) had a more pronounced negative impact on white-patch growth than temperatures of 34 degrees. There were also no major differences in early-embryo survival across your temperature treatments.

I am concerned by the response to my initial suggestion in the first round of review:

“Page 8, Lines 188-191: I assume you excavated and staged the unhatched eggs from the

clutches that were reburied in artificial nests as well. You should provide a summary of these

data also. Or explain why this was not done. If the primary concern of this study is early stage

death I think it is quite important to present the data on which stage of development these

embryos died at." To which the authors state: "This part of the study has now been deleted from the manuscript.”

This part of the study has not been deleted from the manuscript. The authors still state that eggs were reburied in artificial nests on Mon Repos Beach. Is there excavation data with data on stage of death for the eggs? Given the central concern of the study is early stage death, it is important that data on the stage that embryos died at is reported in as much detail as possible. Or provide explanation why this was not done.

Reviewer 2 brings up a similar point which could be answered if the hatching success and stage of embryonic death data is supplied for the eggs that were reburied in artificial nests at Mon Repos:

“L261-263 Again, based on the presented data I’d like to know whether early hypoxia or

hypercapnia had an effect on late embryo mortality in ambient gas tensions. Did the authors

test for such an effect? Whatever the finding, this would be interesting to report in the results.

The data and discussion on late embryo mortality has been deleted from the revised

manuscript.”

These are the only major concerns I still have with the manuscript, the rest below should be straightforward to fix.

Specific points:

Page 2, Line 35: Spacing either side of "/" needs to be fixed.

Page 4, Line 73: "low aggregation nesting aggregation beaches" needs to be changed. Suggest "low density nesting beaches" or "low density nesting beach aggregations".

Page 4, Line 78: Change "with the majority of embryos die at an early stage" to "with the majority of embryos dying at an early stage of development".

Page 5, Lines 106-110: Maybe I've got this wrong. But it seems confusing to me that you ran out of one of the gas mixtures for the treatments for the green turtle experiment when the dates you've reported here suggest this experiment was undertaken first. I suggest double checking your dates for egg collections you have listed in the methods. Or revisit the wording you have used to explain the reason why this treatment was not included in the green turtle experiment.

Page 8, Lines 182-184: Here in the methods, the authors state that two eggs were randomly selected from each treatment (one from each of the "two clutches") to be opened and staged, but further down in the results they state that four eggs from each treatment (one per clutch) were opened and staged. Similarly, at the start of the methods you report four clutches per species being collected at the start of the methods. Furthermore, in the results you report sample numbers of eggs that were opened, 9 and 12 for green turtles and loggerheads respectively. These numbers don’t add up to me, please provide clarification.

Page 9, Line 213: You should present a summary of the data on developmental stage of these opened eggs. The data is not supplied in the raw data file either.

Page 10, Line 221: "all three factors, clutch, temperature and clutch influenced growth" needs to be changed to "all three factors, clutch, temperature and gas influenced growth".

Page 10, Line 226-227: "became increase different" needs to change to "became increasingly different"

Page 15, Line 339: The journal title is abbreviated here whereas other journal titles in the list are not abbreviated. Change for consistency / journal style.

Page 19, Line 435: Change "treatment in for green" to "treatment for green".

Reviewer #2: The authors have done a good job in responding to my previous comments and I see most issues addressed in their revised version. Overall, the manuscript reads much better and is structured in a more intuitive way. Also, the M&M section describes the experiment much more clearly now (largely because it is a simpler experiment). I think, deleting the long-term survival data, which was outside of the proposed study question makes for a simpler more concise paper and avoids much of the previous confusion surrounding the definition of “early stage embryo mortality”. The authors have re-analyzed their data, as suggested, and the new figures convey the major results accurately. To a large extend the discussion adheres more closely to the actual data and much of the previous over-interpretation of the results has been amended. However, in their discussion of the effects of incubation temperature, the authors conclude that elevated temperatures are the likely cause for early embryo mortality, which is not supported by the data presented here. In my mind, this needs to be corrected.

I think the manuscript is much improved from its previous version and I still think the study and it’s findings are interesting and worthwhile publishing. However, I urge the authors to amend the one main comment I have outlined below, before I can recommend the manuscript for publication with PLOS ONE. In addition, I’ve listed a few minor, largely editorial comments below, for the authors to consider (all line numbers refer to those in the revised version with tracked changes)

Main comment

In line L295 and following, it reads that there was no effect of high temperatures on early embryo mortality, but that continued incubation at 34°C caused the death of all sea turtles. As far as I can tell, this is based entirely on the results from previous work. In the present study, embryos incubated at the highest temperature (34°C) had a 97% survival rate. The only adverse temperature effects reported here are based on the observation of one malformed embryo incubated at 33°C. Therefore, it should be stated clearly that the study found no effect of high temperature on early embryo survival (just like there was no effect of adverse gas tensions on early embryo survival). If anything, these results stand in contrast to those of the previous studies cited here and I’d like to see a critical discussion of this discrepancy and its implications.

Again, later in the discussion (L325-328), the authors repeat the conclusion of these high-temperature effects, saying that early stage embryos incubated for a long period at 34°C would experience higher mortality. I find this misleading, because at longer incubations we’re not talking about early embryo mortality anymore, which is the main study question. And again, this is not supported by the present data.

I think the authors should conclude that neither adverse gas tensions or high temperatures had an effect on early embryo survival (at least with the experimental protocol used here) and should rather focus their discussion of the effects of temperature and gases on embryo growth rates, which were significant and are supported by the data. I think these are interesting findings that will inform on the physiological status of the developing embryos, and that are barely discussed in the current version.

Minor comments

Abstract

L19…increase in mortality rate.

L31…but there is evidence suggesting high…I think this is too vague for an abstract. Why not mention what that evidence is?

Introduction

L78 aggregation is repeated

L83 dying

L84 I would clearly define the term “early stage embryo mortality here”. This was done in the abstract, but not the introduction.

M&M

L140 delete “it”

Discussion

L340 Change “In contrast” to “In addition”. The results from the adverse gas exposure and the high temperature were similar in that they both slowed development, but didn’t increase mortality.

L342-348 Again, I urge the authors to discuss the adverse effects of high temperature more critically. The present study found no increased early embryo mortality to speak of and the conclusions here are derived from previous work. I think the authors need to discuss the discrepancy between these findings in more detail. These results are stated correctly in the abstract, but are convoluted in the discussion.

7. PLOS authors have the option to publish the peer review history of their article (what does this mean?). If published, this will include your full peer review and any attached files.

Reviewer #1: **Yes: **Sean Alexander Williamson

Reviewer #2: No

---

## [Author Response · Author response to Decision Letter 1]

22 Oct 2020

Response to reviewers’ comments.

Please thank the reviewers for their feedback and comments and suggestions. Below we detail our response to the reviewer’s comments. Reviewers’ comments are in italics and our responses are in regular font.

Reviewer 1:

General points:

I am concerned that my main criticism of this manuscript has still not been fully addressed. The study aim is to assess the impact of changes in suboptimal gas mixtures and temperatures during early-embryonic development to ascertain whether this could explain high rates of early embryonic death at high-density nesting beaches. Data on stage of embryonic death has not been presented for sufficient comparison of differences between the treatments. Furthermore, in this revised version, there is no longer any substantial mortality data included. This makes it difficult to definitively conclude that any of the treatments applied caused higher rates of early-embryonic death and poorer hatching success in general. The authors did apparently open 2 eggs (or potentially 4?) from each treatment to assess stage of embryonic development at the end of the treatment but this data has not been reported in the manuscript.

Response: The focus of the manuscript is early stage embryonic development, development that occurs within the first 7 days of the eggs being laid. The reason for this highly focused short period of incubation is that at high density nesting beaches, a high proportion of embryo mortality occurs during this period of development. There was virtually no early embryo death detected in any of our treatments during the first 7 days of incubation. However, for the few embryos that did die, we now report the stages that these embryos died at in the revised manuscript. A small sample of eggs were opened at the end of the 7 day treatment period, and all of these embryos were found to be alive. This is now reported in the revised manuscript: “All embryos were alive as indicated by a beating heart when dissected”. 

Given the above, if data on stage of embryonic death cannot be provided, the authors need to temper interpretation of results that temps above 34℃ are more harmful than non-optimal gas concentrations. There was minimal difference in the proportion of embryos that you assumed were still alive at 7 days across all treatments, including temperature treatments. Furthermore, the data you report on relative growth of the white-patch indicate that non-ideal gas mixtures (10%O2, 11%CO2) had a more pronounced negative impact on white-patch growth than temperatures of 34 degrees. There were also no major differences in early-embryo survival across your temperature treatments.

Response: We agree with the reviewer’s comments, but our comments about temperatures above 34oC are based on previously published studies. We have adopted the reviewer’s suggestion and “tempered” our conclusion about high temperature death of early stage embryos, by replacing the would “likely” with “possible” in the revised manuscript: “Hence, high nest temperatures experienced early in incubation is a possible explanation for the large number of nests experiencing a high rate of early embryo death in high-density nesting aggregations.” And we have also tempered our conclusion later in the discussion: “At nesting beaches that experience high-density nesting, the close proximity of nests means that many newly laid clutches could experience high temperatures due to the metabolic heat production of nearby maturing clutches, and this might result in elevated rates of early embryo death, especially if temperatures exceed 34oC for long periods of time.”

I am concerned by the response to my initial suggestion in the first round of review:

“Page 8, Lines 188-191: I assume you excavated and staged the unhatched eggs from the

clutches that were reburied in artificial nests as well. You should provide a summary of these

data also. Or explain why this was not done. If the primary concern of this study is early stage

death I think it is quite important to present the data on which stage of development these

embryos died at. This part of the study has now been deleted from the manuscript.”

This part of the study has not been deleted from the manuscript. The authors still state that eggs were reburied in artificial nests on Mon Repos Beach. Is there excavation data with data on stage of death for the eggs? Given the central concern of the study is early stage death, it is important that data on the stage that embryos died at is reported in as much detail as possible. Or provide explanation why this was not done.

Response: If we had data on hatching success of eggs from each of our treatments, we certainly would have reported it, but we do not have this data. The reviewer has asked why we did not obtain this data. The reason is because of the logistics of obtaining such data. We had limited incubator space and time – we could not continue to incubate our eggs in incubators until they hatched – there was not enough incubator space. Our typical experiment group size for each clutch/treatment combination was 8-10 eggs. As the reviewer knows, such a small number of eggs incubated naturally at nest depth on a beach would not produce enough hatchlings to be able to dig themselves to the surface after they hatch, a group size of 25-30 hatchlings is needed. So, in our study we combined eggs from several clutch/treatments to form new clutches of 100/120 eggs that were incubated on the beach. Because within each of these combined clutches, we could not determine which hatchling came from which treatment, there was no point in evaluating the hatching success of these combined nests. The only reason we stated that eggs were buried to complete incubation on the beach was to assure readers that these eggs were not destroyed after our experimental manipulations. We have now added some further details to the methods section: “We relocated the surviving eggs into artificial nests consisting of 100 to 120 eggs pooled from the different treatments on Mon Repos Beach. Because we could not identify from which treatment hatchlings came from in these artificial nests, we could not assess the effect of treatment on hatching success, so this data is not reported.”

Reviewer 2 brings up a similar point which could be answered if the hatching success and stage of embryonic death data is supplied for the eggs that were reburied in artificial nests at Mon Repos:

“L261-263 Again, based on the presented data I’d like to know whether early hypoxia or

hypercapnia had an effect on late embryo mortality in ambient gas tensions. Did the authors

test for such an effect? Whatever the finding, this would be interesting to report in the results.

The data and discussion on late embryo mortality has been deleted from the revised

manuscript.”

Response: Unfortunately, we cannot make comment on this aspect because we do not know that the hatching success of eggs were. The only data we have on later embryo stage death was from the last two clutches of green turtle eggs incubated. We do not have this data for any of the loggerhead turtle experiments, or the first two clutches of green turtle eggs. So, as suggested by the reviewers in the first round of reviews, presenting this data added confusion to the main aim of assessing temperature and gas mixture effects on early stage death,. It is for this reason that we now do not report the data on later stage death of the last two clutches of green turtle eggs in the revised manuscript.

This is the only major concern I still have with the manuscript; the rest below should be straightforward to fix.

Specific points:

Page 2, Line 35: Spacing either side of "/" needs to be fixed. Response: done.

Page 4, Line 73: "low aggregation nesting aggregation beaches" needs to be changed. Suggest "low density nesting beaches" or "low density nesting beach aggregations". Response: This change has been made. Changed to “low density nesting beaches”.

Page 4, Line 78: Change "with the majority of embryos die at an early stage" to "with the majority of embryos dying at an early stage of development". Response: This change has been made as requested.

Page 5, Lines 106-110: Maybe I've got this wrong. But it seems confusing to me that you ran out of one of the gas mixtures for the treatments for the green turtle experiment when the dates you've reported here suggest this experiment was undertaken first. I suggest double checking your dates for egg collections you have listed in the methods. Or revisit the wording you have used to explain the reason why this treatment was not included in the green turtle experiment. Response: Thanks for pointing this out. The loggerhead turtle eggs were collected in December 2018, not December 2019. This date correction has been made in the revised manuscript.

Page 8, Lines 182-184: Here in the methods, the authors state that two eggs were randomly selected from each treatment (one from each of the "two clutches") to be opened and staged, but further down in the results they state that four eggs from each treatment (one per clutch) were opened and staged. Similarly, at the start of the methods you report four clutches per species being collected at the start of the methods. Furthermore, in the results you report sample numbers of eggs that were opened, 9 and 12 for green turtles and loggerheads respectively. These numbers don’t add up to me, please provide clarification. Response: Thanks for pointing this inconsistency out. We have now corrected this mistake in the revised manuscript.

Page 9, Line 213: You should present a summary of the data on developmental stage of these opened eggs. The data is not supplied in the raw data file either. Response: This data is now reported in a new Table 3 in the revised manuscript.

Page 10, Line 221: "all three factors, clutch, temperature and clutch influenced growth" needs to be changed to "all three factors, clutch, temperature and gas influenced growth". Response: This change has been made.

Page 10, Line 226-227: "became increase different" needs to change to "became increasingly different" Response: This change has been made.

Page 15, Line 339: The journal title is abbreviated here whereas other journal titles in the list are not abbreviated. Change for consistency / journal style. Response: This change has been made.

Page 19, Line 435: Change "treatment in for green" to "treatment for green". Response: This change has been made.

---

## [Decision Letter · Decision Letter 2]

17 Nov 2020

The effect of respiratory gases and incubation temperature on early stage embryonic development in sea turtles

PONE-D-20-13033R2

Dear Dr. Booth,

We’re pleased to inform you that your manuscript has been judged scientifically suitable for publication and will be formally accepted for publication once it meets all outstanding technical requirements.

Kind regards,

Frank Melzner

Academic Editor

PLOS ONE

Additional Editor Comments (optional):

Reviewers' comments:

Reviewer's Responses to Questions

**Comments to the Author**

1. If the authors have adequately addressed your comments raised in a previous round of review and you feel that this manuscript is now acceptable for publication, you may indicate that here to bypass the “Comments to the Author” section, enter your conflict of interest statement in the “Confidential to Editor” section, and submit your "Accept" recommendation.

Reviewer #1: All comments have been addressed

Reviewer #2: All comments have been addressed

2. Is the manuscript technically sound, and do the data support the conclusions?

Reviewer #1: Yes

Reviewer #2: Yes

3. Has the statistical analysis been performed appropriately and rigorously? 

Reviewer #1: Yes

Reviewer #2: Yes

4. Have the authors made all data underlying the findings in their manuscript fully available?

Reviewer #1: Yes

Reviewer #2: Yes

5. Is the manuscript presented in an intelligible fashion and written in standard English?

Reviewer #1: Yes

Reviewer #2: Yes

6. Review Comments to the Author

Reviewer #1: I am satisfied that the authors have made sufficient changes to the manuscript to address my previous concerns. I congratulate the authors again on completing an important and needed study. I believe that this article should now be accepted for publication in PLOS One.

I don’t have any further suggestions for changes or edits to the manuscript. My only comment is, for future studies, the authors should consider marking individual eggs with pencil in multiple locations prior to reburial when logistics, such as limited incubator space and time, require that eggs be reburied in nests with eggs from different treatments or females mixed together. It is not guaranteed to work, but provided the eggshells haven’t degraded too much when you excavate the nest you can identify which individual eggs hatched and which did not.

Reviewer #2: The authors have done a great job in addressing my previous comments and I see all my major points addressed in this revised version of their manuscript. The discussion and main conclusions follow nicely from the available data and there is a clear “red line” throughout the paper. I only have two minor suggestions for the authors to consider as they see fit, but have no further comments.

L204 and 209 The authors refer to Miller’s developmental stages throughout this section of the results, a metric which has not been introduced previously in the manuscript. For the naïve reader, like myself, it may be worthwhile describing the method in a sentence and mention what it’s based on and the range it uses (e.g. 0 is oviposition and 15 or so is hatch?).

One of the major conclusions of the paper is that warmer nesting temperatures are more likely to decreasing hatching success compared to adverse gas conditions. The authors talk about the effect of metabolically produced heat in high nesting areas that could cause such increased mortality rates, but not whether rising temperatures in the environment are accelerating these trends. Do the authors have any evidence that average nesting temperatures have increased due to climate change? If such data exist it may be worthwhile mentioning this in a sentence or two, perhaps in the last section on management strategies and perspectives. In my mind, this would be a nice way to round up the study and raise awareness about the need for conservation strategies, which would further strengthen the paper.

7. PLOS authors have the option to publish the peer review history of their article (what does this mean?). If published, this will include your full peer review and any attached files.

Reviewer #1: **Yes: **Sean Alexander Williamson

Reviewer #2: No

---

## [Editor Report · Acceptance letter]

20 Nov 2020

PONE-D-20-13033R2 

The effect of respiratory gases and incubation temperature on early stage embryonic development in sea turtles. 

Dear Dr. Booth:

I'm pleased to inform you that your manuscript has been deemed suitable for publication in PLOS ONE. Congratulations! Your manuscript is now with our production department. 

Kind regards, 

on behalf of

Dr. Frank Melzner 

Academic Editor

PLOS ONE